# Direct Powder Extrusion of Paracetamol Loaded Mixtures for 3D Printed Pharmaceutics for Personalized Medicine via Low Temperature Thermal Processing

**DOI:** 10.3390/pharmaceutics13060907

**Published:** 2021-06-19

**Authors:** Xabier Mendibil, Gaizka Tena, Alaine Duque, Nerea Uranga, Miguel Ángel Campanero, Jesús Alonso

**Affiliations:** 1Tekniker, Basque Research and Technology Alliance (BRTA), Iñaki Goenaga N5, 20600 Eibar, Spain; xabier.mendibil@tekniker.es (X.M.); gaizka.tena@tekniker.es (G.T.); nerea.uranga@tekniker.es (N.U.); 2A3Z Advanced Analytical Consulting Services, Astondo bidea S/N, 48160 Derio, Spain; alainedcrr@gmail.com (A.D.); macampanero@a3zadvanced.com (M.Á.C.)

**Keywords:** 3D printing, personalized medicine, hot melt extrusion, starch, paracetamol, hydroxypropyl cellulose

## Abstract

Three-dimensional printed drug development is nowadays an active area in the pharmaceutical industry, where the search for an appropriate edible carrier that permits the thermal processing of the mixture at temperature levels that are safe for the drug is an important field of study. Here, potato starch and hydroxypropyl cellulose based mixtures loaded with paracetamol up to 50% in weight were processed by hot melt extrusion at 85 °C to test their suitability to be thermally processed. The extruded mixtures were tested by liquid chromatography to analyze their release curves and were thermally characterized. The drug recovery was observed to be highly dependent on the initial moisture level of the mixture, the samples being prepared with an addition of water at a ratio of 3% in weight proportional to the starch amount, highly soluble and easy to extrude. The release curves showed a slow and steady drug liberation compared to a commercially available paracetamol tablet, reaching the 100% of recovery at 60 min. The samples aged for 6 weeks showed slower drug release curves compared to fresh samples, this effect being attributable to the loss of moisture. The paracetamol loaded mixture in powder form was used to print pills with different sizes and geometries in a fused deposition modelling three-dimensional printer modified with a commercially available powder extrusion head, showing the potential of this formulation for use in personalized medicine.

## 1. Introduction

Medication adherence is crucial to achieve a properly successful therapy and the discontinuation of the treatment involves high economic costs and several thousands of premature deaths each year [1,2,3,4]. It is estimated that up to 50% of medicines are not used as prescribed; this misuse can happen for diverse reasons: confusion caused by polypharmacy, misunderstanding of the treatment, lack of health knowledge, poor palatability, etc. [5,6]. Lately, 3D Printed (3DP) drugs have drawn attention as an alternative to improve the adherence to treatment, improving the organoleptic characteristics and facilitating identification by using different shapes and colors [7,8,9]. The use of 3D printing allows the personalization of the oral forms (3DP pills) not only to improve adherence, but to customize the drug dosage, which is of great interest in the definition of the prescription for adults and is critical in pediatrics [10,11,12,13].

Different 3D printing technologies have been described in the literature to prepare 3DP pills [14,15,16,17]. Fused deposition modelling (FDM) and Semi Solid Extrusion (SSE) are the most straightforward technologies to prepare different mixtures and combinations of drugs and excipients, where the active pharmaceutical ingredient (API) is directly blended with or impregnated in the excipient/carrier to subsequently be processed. Seoane-Viaño et al. [18] did a thorough review of SSE, highlighting the numerous applications this technology has and discussing some of its drawbacks: post-printing drying process, complete solvent evaporation, risk of material collapse and loss of shape. On the contrary, FDM does not show those post processing problems, but it is limited by the low number of Food and Drug Administration (FDA) approved thermoplastic polymers and the thermal degradation the drug undergoes during the process [14,19,20]. Nevertheless, FDM has become one of the most popular 3D printing technologies, offering consumer level use at reduced investment [21,22]. The filament extrusion is the most extended FDM printing technique and there are commercially available solutions that upgrade the printer head to allow processing polymers in pellet and powder form [23], avoiding the need of filament production and reducing in this way the thermal stress of the deposited material.

Most of the found literature, where the FDM is used to prepare 3DP pills, applies the loaded filament approach; some authors use the impregnation method, where the drug is added to a preexisting polymer filament [24,25], but the most settled method is the preparation of the loaded filament by thermal processing. In this procedure, the drug loaded filament is manufactured by hot melt extrusion to later be fed into the 3D printer and be heated again. For example, Đuranović et al. [26] processed paracetamol loaded filaments based on poly(ε-caprolactone) (PCL) and poly ether oxide (PEO) polymers. They succeeded in extruding filaments and printing with drug loads up to 48% wt at 130 °C but their polymer–drug blends underwent two thermal processes, and although the processing temperature was under the melting point of the paracetamol, their samples showed yellowish discoloration, indicating some degradation or oxidation of the components. Gorkem Buyukgoz et al. [27] extruded filament based on Hydroxypropyl Cellulose (HPC) loaded with Griseofulvin at 140 °C to subsequently 3D print it at 170 °C. Other examples of drug loaded filament preparation to feed the 3D printer can be found in [28,29,30,31,32].

By contrast, Goyanes et al. [23] used a FDM modified printer to directly 3D print from powder form, avoiding in this way the need of filament preparation. They tested different molecular weight (Mw) HPC loaded with 35% itraconazole and successfully printed pills processing the blends at 170 °C. Ong et al. [33], following the work made by Goyanes, also tested the direct printing of powder mixtures of different HPC grades loaded with PEO and tramadol as API, achieving successful 3DP pills processed at 170 °C. The use of modified 3D printers to directly print the drug and excipient in powder form, reducing in this way the thermal stress on the API, seems to be the appropriate process chain to success in 3DP drug development. The search of suitable edible low temperature processing excipients to reduce the need of high temperatures in the printer nozzle is the next step to foster the research in this area. The main drawback in the 3DP drug development is the need of an appropriate drug carrier that allows the thermal processing of the mixture, and that meets the requirements of being edible, processable at low temperature and with customizable solubility to control the release curve.

In this regard, starch and HPC arise as interesting excipients as they can be processed at low temperatures and are widely used in pharma technology to prepare pharmaceutical formulations and are also extensively used in cosmetics and food products. The Guar gum (Gg) is used as binder and disintegrant in solid dosages and in low percentages can ease the thermal processing of the mixture of excipients [34,35].

Paracetamol is authorized to be used to alleviate mild to moderate pain caused by headaches, toothache, sprain or strains. This drug is available in different dosage oral solid forms such as tables, capsules, suspensions. The active dose ranges from 300 mg to 1000 mg. Direct compression is the most common method used for tableting production [36]. The direct compression is a pharmaceutical operation limited to relatively low drug loading tablet production due to the mechanical properties of some active pharmacological ingredient compromising compression characteristics. For paracetamol tables, only 30–40% wt of the active can be accommodated. This can cause patient non-compliance due to the large tablets [36]. Moreover, the crystals of paracetamol display low flowability and poor compression ability, and when such crystals are compressed into tablets, they show massive elastic deformation under pressure and a tendency to cause problems with tablets, such as chipping, capping, stress cracking, lamination, ticking and picking [37].

The present paper studies the suitability of different formulations, based on starch and HPC with different proportions of acetaminophen (paracetamol), to be processed at a low temperature by hot melt extrusion, searching the compliance to be printed in an FDM 3D printer. The different drug excipient mixtures were thermally processed, with different quantities of Gg, by hot melt extrusion (HME). The dissolution rates and drug release curves of each combination were measured, and the samples were thermally characterized to ensure that the API did not undergo thermal degradation. Finally, 3DP pills were obtained directly from powder form to show the potential of those starch/HPC based mixtures to be processed by means of FDM 3D printing at low temperatures.

## 2. Materials and Methods

### 2.1. Materials, Reagents and Reference Products

Potato starch, HPC and paracetamol (acetaminophen 98%) were purchased from Sigma Aldrich (Darmstadt, Germany). Guar gum from HSN was purchased directly to the vendor. Hydrochloric acid (25%) for analysis (*w*/*v*) and HPLC grade acetonitrile were obtained from Thermo Fisher Scientific (Goteborg, Sweden), and sodium phosphate monobasic monohydrate used for the mobile phase preparation was purchased from Sigma Aldrich (≥99%, Darmstadt, Germany).

Ultrapure water used in the experiments was obtained from Milli-Q water purification system (Merck Millipore, Darmstadt, Germany).

Paracetamol 650 mg tablet from Teva Pharma, S.L.U. (Madrid, Spain) was used as control in release assays.

Paracetamol (Sigma Aldrich, Darmstadt, Germany) was also used as reference product in HPLC analysis.

### 2.2. Equipment

A precision analytical balance from Mettler Toledo (Columbus, OH, USA) and a twin conical screw Haake Minilab II micro extruder from Thermo Fisher Scientific (Goteborg, Sweden) were employed for the preparation of the different formulations.

Differential Scanning Calorimetry (DSC) and Thermo Gravimetric Analysis (TGA) employed in the thermal characterization of the samples were carried out in a GC 200 Star System and in a TGA/DSC 1 LF from Mettler Toledo (Columbus, OH, USA).

Gilson micropipettes from Thermo Fisher Scientific (Goteborg, Sweden) were used in the preparation of reagents, mobile phase, test and reference samples. A precision analytical balance GRAM FV-220 from Gram (Kolding, Denmark) and magnetic stirrers SH-2 from Anzeser (Guangdong, China) were used for the reagent preparation. Samples were filtered across 0.45 µm PTFE (polytetrafluoroethylene) filters from Pall Corporation (Ilfracombe, UK) before the chromatographic analysis. Vials sealed with 11 mm caps from Agilent Technologies (Waldbronn, Germany) were used.

HPLC analysis was performed on an Agilent 1100 series HPLC system equipped with a G1313A autosampler, a G1311A quaternary pump and a G1314A variable wavelength detector (VWR). The software used was Agilent ChemStation B.01.03. The chromatographic column was a Zorbax Eclipse XDB-C18 (150 mm × 4.6 mm, 5 µm particle size) purchased to Agilent Technologies (Waldbronn, Germany).

### 2.3. HPLC Analysis

The chromatographic method was carried out with a mobile phase program of 93.3% 0.0125 M sodium phosphate monobasic monohydrate in ultrapure water (pH = 4.95) filtered across a 0.45 µm PVDF (polyvinylidene fluoride) filter (Merck Millipore, Darmstadt, Germany) and 6.7% acetonitrile with a total run of 10 min. The injection volume was 5 μL and the detection wavelength was set at 254 nm. The chromatographic column was maintained at 30 °C with a flow rate of 1 mL/min.

A 0.5 mg/mL standard solution of paracetamol in 0.1 N hydrochloric acid was prepared for sample quantification. The sample concentrations were calculated by using the response factor (RF) obtained from this standard. The acceptance criterion for each analytical batch was coefficient of variation (CV) ≤ 5%.

### 2.4. Processing and Characterization

Four different paracetamol proportions and three Gg quantities were studied yielding twelve mixture combinations. The defined paracetamol proportions were: 5, 20, 35 and 50 percentage in weight (% wt) and 0 (zero), 5 and 10% wt in the case of the Gg. The quantity of the starch added to the mixture was varied considering the paracetamol and Gg present in each sample to maintain a constant total dry mass of 15 g for each combination. Ultrapure water was added in 3% wt ratio proportional to the amount of starch of the sample; this implied that the quantity of water varied for each combination accordingly to the starch amount. The total HPC was kept constant at 25% wt for all the combinations. The preliminary TGA of each constituent showed that the starch and the Gg had a not negligible moisture level (data in Appendix A). A summary of the prepared samples and their composition can be found in Table 1, showing the quantities of each constituent and the calculated percentage of water present in the mixture, taking into account the added ultrapure water and the moisture of the constituents.

The constituents of each sample were measured by means of a precision analytical balance and stirred vigorously with a stainless-steel spoon in a glass container prior to adding the ultrapure water. The water was added to the mixture of powders while the overall weight of the wet mixture was controlled with the balance. Once the water was added, it was stirred again to moisten the mixture evenly and prevent clumps and aggregates. The wet powder mixture was directly put into the hopper of the extruder.

The extrusion of the API loaded mixtures was carried out using a twin conical screw micro extruder. The processing conditions were kept constant for all the different mixtures, setting the extruder temperature at 85 °C and the screw speed at 20 rpm. The extrudate was not pulled and it was let freely flow over a stainless-steel surface, which was at room temperature, at the outlet of the nozzle of the extruder.

The samples were extruded in two batches: the first batch was used to perform a screening where the solubility of the prepared mixtures was tested. In this screening, all the combinations were analyzed to determine which sample group, with a given Gg ratio, showed a better compromise between dissolution and processability. With the selected Gg proportion, a second batch was prepared. The second batch was characterized in fresh and aged conditions to check the influence of the ageing of samples. The samples were aged at room temperature (RT) in sealed bags and kept in a dark and dry place. The samples were characterized via release curve tests and TGA and DSC thermal tests.

### 2.5. Release Studies

#### 2.5.1. Assay

The samples of the formulations (*n* = 3) were cut in different measures and diluted to a 0.5 mg/mL paracetamol equivalent concentration in 0.1 N hydrochloric acid. These mixtures were transferred into a water bath at 37 °C under stirring. After 30 min, a 1 mL sample was taken, filtered with a 0.45 µm PTFE filter, transferred to a chromatography vial and analyzed by the HPLC method detailed in Section 2.3. Finally, the paracetamol released concentration at 30 min was quantified.

#### 2.5.2. Dissolution Studies

The samples of the four formulations (*n* = 3) cut in different measures and the commercial paracetamol tablet (*n* = 3) used as control were diluted to a 0.5 mg/mL paracetamol equivalent concentration in 0.1 N hydrochloric acid. These samples were transferred into a water bath at 37 °C under stirring. At the times of 0, 5, 10, 15, 30 and 60 min, a 1 mL sample was taken, filtered with a 0.45 µm PTFE filter, transferred to a chromatography vial and analyzed by the HPLC method detailed in Section 2.3. In the case of the commercial paracetamol tablet, the assay was stopped at 30 min because of its early complete dissolution. Finally, the paracetamol released concentration at each time was quantified and the released concentration was plotted versus time.

### 2.6. Thermal Characterization

The samples were thermally characterized by means of DSC and TGA, and a single test per sample was performed. The DSC was carried out by performing a ramp from 30 °C to 250 °C at 10 °C/min, N_2_ flow 50 mL/min. The TGA was carried out by performing a ramp from 30 °C to 500 °C at 10 °C/min, air flow 50 mL/min. Samples of the four formulations prepared in the second batch were thermally characterized in aged condition, and additionally, samples of formulations containing 20 and 50% wt of paracetamol were also characterized in fresh conditions to compare with the aged ones.

### 2.7. 3D Print

A TEVO Michelangelo 3D (Guandong, China) printer modified with powder extruder head from MahorXYZ (Pamplona, Spain) kindly provided by DOMOTEK (Tolosa, Spain) was used for printing pill demonstrators to test the suitability of the characterized mixtures. The geometry was sliced using Ultimaker CURA (Utrecht, The Netherlands) software and printed at 90 °C range. The slicing and printing conditions were: layer height 0.1 mm, wall thickness 0.8 mm, top/bottom layers 3, infill 100%, concentric pattern, flow 150%, bed temperature 75 °C, nozzle temperature 90 °C, print speed 25 mm/s, wall speed 12.5 mm/s, retraction on.

## 3. Results

### 3.1. Extrusion of the Samples

The prepared mixtures were extruded in rectangular shape stripes, copying the geometry and dimensions of the outlet of the extruder (4 × 1 mm). The extruded stripes were whiteish and opaque, except for the samples containing only 5% wt of paracetamol, which showed slightly translucent appearance. A picture of a mixture with formulation containing 50% wt of API and 5% wt of Gg is shown in Figure 1. All the mixtures happened to be highly flexible and sticky when warm, in particular, the blends with 5% wt of Gg. Nearly 50 cm was successfully extruded for each mixture.

### 3.2. Solubility and Release Curves

#### 3.2.1. Preliminary Screening

The extruded stripes of all formulations were dissolved almost completely after 30 min under agitation at 37 °C. Three different samples of each formulation were tested, and the mean values of the recovery rates were plotted as shown in Figure 2 (data in Appendix A). All mixtures showed recovery values over 70%, the sample with the worst recovery being the mixture containing 10% wt Gg and 50% wt of paracetamol, yielding (72 ± 11%). The samples with 35% wt of paracetamol showed low values of recovery (80 ± 9%; 79 ± 5% and 81 ± 6%) independent of the amount of Gg. All the formulations containing 10% wt of Gg showed recovery values under 90%, indicating some negative effect in the dissolution caused by the presence of the gum. The samples containing 0 and 5% wt of Gg, except of those of 35% wt of paracetamol, showed recovery values over 85%, exhibiting a remarkable dissolution rate even for the mixtures containing 50% wt of paracetamol (87 ± 4% and 96 ± 7%). The samples containing 0 and 5% wt of Gg and lower quantities of paracetamol (5 and 20% wt) showed the highest recovery rates (103 ± 2%, 98 ± 5%, 104 ± 2% and 96 ± 2%).

#### 3.2.2. Dissolution Studies

The mixtures containing the 5% wt of Gg were selected because of their high solubility and ease processing to continue the characterization. A second batch was extruded to dispose of enough samples for testing and they were aged for 6 weeks at RT. For each paracetamol proportion, three different segments of the same batch of each extruded strip were tested. The results of the release assay described in Section 2.5.2 are collected in Appendix A and are plotted in Figure 3a, and a detail of the recovery values at times 30 and 60 min on the right panel.

At minute 0, all the samples, including the paracetamol tablet, showed a burst release of 5%, the mixture with 5% wt paracetamol showing the highest value of (4.7 ± 1.2%). The paracetamol release was slower in the extruded mixtures than the paracetamol tablet. At minute 5, the paracetamol tablet showed a pronounced rise to (78 ± 8%) while the extruded mixtures kept values in the range 23 to 18%, the highest value being 23% corresponding to the mixture with 5% wt of API and the lowest value of (18 ± 2%) to the mixture containing the 50% wt. At minute 10, the paracetamol pill reached almost its maximum with a recovery rate of (98 ± 1%) which evolved to (99 ± 1%) at minute 30. All the mixtures showed a steady increment, showing the maximum recovery rates at minute 60. The mixture with 5% wt of paracetamol showed the highest values among the extruded samples growing progressively from minute 10 to 60 with values of (37 ± 1%, 49± 3%, 83 ± 6% and 109 ± 1%), whereas the samples with 50% wt of paracetamol yielded the lowest values (30 ± 1%, 40 ± 4%, 67 ± 2% and 95 ± 3%). For all the tested samplings the recovery rates showed a slight reduction inversely proportional to the amount of API, being the recovery rate of the heavily loaded samples lower than those samples containing lower amounts of API. The samples containing 20 and 35% wt of paracetamol showed maximum values of (102 ± 1%) and (104 ± 1%) at minute 60. On Figure 3b it can be observed that the mixture containing 50% wt of paracetamol showed significant differences (*p* < 0.05) to the other formulations at 30 and 60 min, while the rest of the samples did not show significant differences (*p* > 0.05) among them at minute 30. At minute 60, the mixture containing only 5% wt was also observed to be significantly different to the rest of the mixtures at the same time as the samples containing 20 and 35% wt did not shown statistically relevant differences among them at minute 60.

#### 3.2.3. Ageing of the Samples

The release of fresh and aged samples was compared at 30 min under agitation at 37 °C. The results are shown in Figure 4 and collected in Appendix A. Fresh samples yielded higher recovery values compared to aged ones for all API proportions. Fresh samples showed a clear rising trend proportional to the API content, while the aged samples showed the opposite tendency. The observed recovery percentages of fresh samples were (88 ± 3%, 90 ± 1%, 93 ± 2% and 98 ± 3%) and for aged samples were (80 ± 1%, 73 ± 2%, 74 ± 3% and 74 ± 3%) for each paracetamol proportion of 5, 20, 35 and 50% wt.

### 3.3. Thermal Characterization

Samples of mixtures containing 5% wt of Gg and 20% wt and 50% wt of paracetamol were thermally characterized in fresh and aged conditions. No remarkable crystallization differences between aged and fresh samples were observed; the results are shown in Appendix A and the DSC curves of each constituent are shown in Appendix A. The DSC of all the aged samples is shown in Figure 5a, where four curves are shown corresponding to each tested sample. The sample containing 50% wt of API showed a narrow endothermic peak centered at 170 °C, corresponding to the melting temperature of the paracetamol in a crystalline form, and no other clear thermic events were detectable. The curve corresponding to the formulation with 35% wt of API showed a complex curve with the center of the peak corresponding to the paracetamol slightly sifted to 165 °C, and it showed a wide endothermic peak which started at 110 °C and was centered at 140 °C with two small spikes around 125 °C and 145 °C. The curve corresponding to the formulation with 20% wt and 5% wt of paracetamol showed curves with similar behavior, a wide peak centered at temperatures lower than 140 °C and showing a peak corresponding to the paracetamol less noticeable as the ratio of API decreases and is shifted to lower temperatures; in those curves the small peaks that appeared in the 35% wt curve are also subtly present.

The TGA analysis of the samples (Figure 5b) showed similar results for all the tested formulations. Samples containing 5 and 20% wt of API showed a small mass loss beginning at 100 °C, while for the mixtures with 35 and 50% wt of paracetamol, this step is less accentuated. All the samples started to lose mass significantly at 250 °C and above, showing a slight change in the slope around 300 to 350 °C and stagnating at 375 °C. Most of the mass is lost between 250 °C and 375 °C, but the most intense mass loss occurs between 300–350 °C where a change in slope is observed.

### 3.4. 3 DP Pills

Different geometries were printed to demonstrate the suitability of the characterized paracetamol and starch based mixture to print pill like geometries of the desired shape and size. In Figure 6a, a detail of the printer head of the 3D printer is shown printing a cylindrical tablet and in Figure 6b some printed pills are shown. The formulation of the shown pills was the mixture of 5% wt Gg and 20% wt paracetamol.

## 4. Discussion

The inclusion of a 3D printer in the drug manufacturing process could eliminate a number of steps in the solid dosage pipeline production process, such as powder milling, wet granulation, dry granulation, tablet compression, coating and long-term stability studying test, especially, when there is a limited quantity of active ingredients at an early drug development stage [38].

Starch is a polysaccharide of glucose composed of α-amylose and amylopectin polymers and it is one of the most used excipients in pharmacology [39,40,41]. The melting temperature depends on the ratio of α-amylose/amylopectin, Mw and water content and, in the native dry form of the starch, it exceeds the degradation temperature, making it not appropriate to be processed by thermomechanical techniques [42,43]. However, the control of the moisture level in the starch allows one to process it like a thermoplastic. During the heating, the water firstly enters the amorphous regions of the starch in a reversible way, plasticizing the hydrated mixture and allowing its thermal processing [44,45]. The control of the water quantity in the starch permits one to control and/or prevent the gelatinization and phase transitions, permitting one to process the starch by means of hot melt extrusion at reasonably low temperatures of less than 100 °C. This relatively low processing temperature makes the starch a proper excipient to be mixed with APIs and be used in FDM of 3DP medicines [39,46,47].

HPC is a non-toxic and non-irritant polysaccharide that is widely used as a binder in oral solid dosage forms [48,49]. It is an amorphous polymer which shows a softening temperature range between 100 °C and 150 °C and it has been previously used to prepare thermally processed drug mixtures, indicating that it is an excipient prone to be used in FDM to manufacture 3DP medicaments [29,50,51,52].

Guar Gum is a polysaccharide of galactose and mannose extracted from the seed of the *Cyamopsis tetragonoloba*. It is widely used in the food industry because of its gumming and pasting properties in combination with starch based materials [35,53].

All the tested mixtures were easily extruded at 85 °C, even those containing 50% wt of paracetamol, indicating that higher concentrations of API could be processable, as observed by Bialleck et al. [47], who achieved the pelletization of starch based mixtures loaded with APIs up to 80% wt. The use of a blend of starch and HPC with the paracetamol resulted in a combination easy to process at lower temperatures than those identified in the literature, which are above 160 °C for blends with HPC processed by FDM [23,27,29] and in the same level of temperatures (70–80 °C) needed to process starch based blends processed by SSE [54,55].

The addition of Gg in low quantities did not noticeably affect the extrusion process, the samples containing Gg being slightly stickier at the outlet of the extruder, which could be of interest to improve the FDM process. The characterization of the expansion of the extrudate and analysis of the viscosity of the mixtures containing Gg and starch fall outside the scope of the present study and can be found extensively discussed in the literature [56,57,58,59].

The presence of Gg in a proportion of 10% wt did impact on the recovery percentage, notably decreasing the recovery values (Figure 2). This negative effect on the recovery caused by the addition of Gg was also described by Saini et al. [60], who observed that the solubility of an API loaded mixture could be decreased over a specific concentration of Gg because of adsorption phenomena. Mixtures without and with a low percentage of Gg (5% wt) (white and solid grey columns in Figure 2) did not show statistically relevant differences (*p* > 0.05), showing similar recovery rate values, indicating that the low amount of Gg did not affect the recovery rate. Surprisingly, both 0 and 5% wt Gg content mixtures showed notable low recovery values for the formulations containing 35% wt of paracetamol. This behavior was not observed for the second batch used to measure the extended release curve, indicating that this misbehavior could be related to human error in the preparation of the samples or weighing of the constituents. A decreasing trend was observed for all the Gg formulations with the increment of paracetamol. This effect could be caused by the reduction of the amount of starch in the formulations, which is one of the main constituents in the disintegration process, taking into account that the proportion of HPC was kept constant for all the tested mixtures.

The second batch of the prepared samples showed similar dissolution behavior and recovery rates (Figure 3a) to the first batch set for the preliminary screening. In this case, the formulation with 35% wt of paracetamol did not show anomalous values and followed the same recovery trend as the other concentrations of API, indicating that the observed low recovery value of the mixture with 35% wt of the first batch could be related to some preparation error and not to a synergic effect caused by the ratio of the constituents. The prepared extruded stripes showed a slow steady dissolution over time compared to the paracetamol tablet, which completely dissolved in the first 10 min, while the extruded mixtures needed up to 60 min to release the API completely to the media. The measurement of the recovery value showed a high reproducibility, with CV values lower than 7%. The mixtures showed a clear decreasing trend with the increment of API in the formulation (Figure 3b). This trend was also observed in the first batch used in the screening test and it is attributed to the diminution of the amount of starch in the mixture.

The tested samples were paralepidids copying the geometry of the outlet of the extruder and did not present any tailored dimensions, porosity or surface to volume ratio that can be easily controllable by means of 3DP tablet design. Nevertheless, they exhibited slow drug release compared to the commercial tablet, but very fast compared to other formulations found in the literature [23,24,27,61]. This release rate can be controlled by designing the tablet geometry [62,63] or by modifying drug carrier matrix materials [64]. Having a formulation with a smooth release rate is of great interest for future research.

The recovery percentage was observed to be strongly affected by the ageing condition of the samples. In the release test at 30 min (results shown in Figure 4 and Appendix A), the aged samples were observed to behave similarly to the samples tested in the extended release, yielding smaller percental recovery values as the amount of API present in the sample increased and the quantity of starch was reduced. In the case of fresh extruded samples, the recovery trend was observed to be counter-wise, the most API loaded samples being those which gave higher recovery values. Those aged samples did not show any crystallization compared with their fresh counterparts (Appendix A, left panel) and the observed variations seem to be related to the moisture content of the sample, the aged samples having lower humidity content (Appendix A, right panel). Aviara et al. [45] observed variation in the solubility of sorghum starch depending on its moisture level and drying process, describing that samples having lower humidity showed lower solubility. The samples used in the extended release curve test shown in Figure 3 were also aged samples which showed recovery values of around 100% at 60 min. In the case of the recovery test, at a single point at 30 min, those samples showed lower recovery values compared to their fresh counterparts, only indicating that the aged samples need more time to reach the maximum recovery value. This delay in the release can be attributed to the water penetration in the aged (drier) sample matrix to start disintegrating it.

The DSC of the tested samples showed complex curves where all the constituents were mostly recognizable, depending on their proportion in the formulation of the sample and if not masked by other, stronger thermal events of another constituent. The samples were extruded at 85 °C, which is far lower than any observed thermal event in the DSC; that could involve that no thermal stress has been applied to any of the constituents. The small spikes around 125 °C and 140 °C correspond to the melting temperatures of the HPC and the Gg as stated in the literature [52,65]. The peak of the paracetamol centered at 170 °C was clearly identifiable in the sample with 50% wt of API, and it disappears in the sample with only 5% wt, where the most remarked wide endothermic curve corresponds to the potato starch endotherm M1 centered at 120 °C [44,46,66], which in turn does not appear in the curve of the sample with 50% wt, where the ratio of starch is smaller. The absence of the endothermic peak of the paracetamol in the low drug concentration samples, can be related to a masking effect, the endotherm of the starch being the most prominent and noticeable event; but it could also be related to some interaction with the Gg, which was set at 5% wt in those thermally tested mixtures, and whose interaction effect would be more noticeable in those samples with low drug presence. This interaction with the Gg would be reversible. As observed in the chromatography assay of the dilution test samples, the retention time of the paracetamol chromatographic peak was the same as that of the corresponding reference product. No physicochemical interaction between paracetamol and the excipients was denoted. The absence of a clear endothermic peak around 60 °C suggests that no starch gelatinization occurred during the heating cycle in the DSC test, but it could have happened during the extrusion and not appeared in the following heating processes, as observed by Lin et al. [67]. However, the gelatinization of the samples is not expected to have happened because of the low water proportion used in the formulations and the high dissolution all the formulations shown [44]. In a preliminary formulation test that was carried out with higher water ratios (Appendix A), where the amount of water was proportional to the overall excipient amount instead of to only the starch amount, formulations with 35% wt and 50% wt of API, where the quantities of starch were the lowest, showed no dissolution nor paracetamol recovery, indicating that some degree of starch gelatinization occurred; those formulations were discarded.

The observed step starting at 100 °C in the TGA analysis (Figure 5b) corresponds to the moisture content of the samples, the samples with a higher proportion of starch (5% wt and 20% wt of API) being those which showed remarked steps of weight loss. The observed mass losses that are attributable to the moisture of the samples are in accordance with the calculated moisture percentage (Table 1), indicating that the amount of water in the samples was not remarkably varied during the process. The samples did not show other remarkable differences at higher temperatures and behaved similarly.

The 3DP pills were printed with 20% wt of paracetamol as an example of the suitability of the characterized mixture to be printed with 3D printers. Besides the material showing some adherence problems to the printer table, it had a good printing performance, showing that the selected formulations, added to the fact of their good dissolution performance and recovery values, are appropriate to be used in personalized drug dosages and to improve the adherence to treatment.

## 5. Conclusions

Starch and HPC based mixtures loaded with paracetamol were processed by hot melt extrusion. The samples prepared with the addition of water at 3% wt ratio proportional to the starch amount turned out to be easily extrudable at low temperature (85 °C) and highly soluble, yielding recovery rates up to 100% at 60 min in HCl 0.1 N at 37 °C. The addition of Gg in 5% wt proportion eased the extrusion and did not affect the recovery values while the recovery dropped for samples containing 10% wt of Gg. The samples aged for 6 weeks showed recovery values dependent on the amount of paracetamol, the samples with higher drug content being those which showed lower recovery values. The fresh samples showed faster drug release comparing to the aged ones, showing recovery values close to 90% at 30 min, and showing recovery rates proportional to the paracetamol amount in contrast to the aged ones. The proposed excipient drug mixtures were used in a regular FDM 3D printer, to which a commercially available powder extruder head had been added. Different geometry pills were printed, demonstrating the suitability of this blend for personalized drug development at low temperature and without the need of specialized equipment, paving the way for new development and studies with thermal sensitive drugs. The analysis of the effect of the printed pill geometry and the printing strategies on the release curve is planned as future work.

## Figures and Tables

**Figure 1 pharmaceutics-13-00907-f001:**
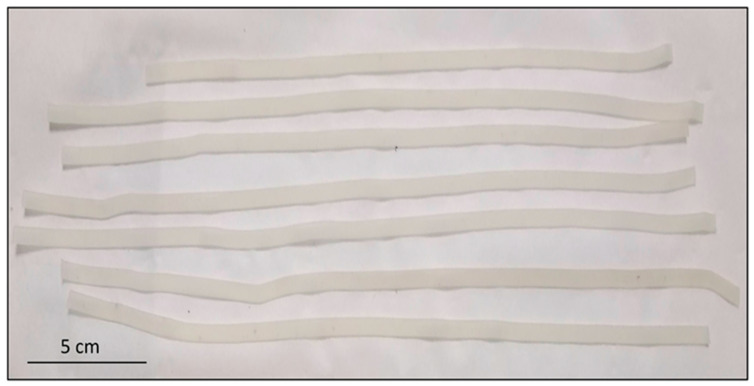
Picture showing the extruded stripes of the batch containing 50% wt of API and 5% wt of Guar gum. Scale bar 5 cm.

**Figure 2 pharmaceutics-13-00907-f002:**
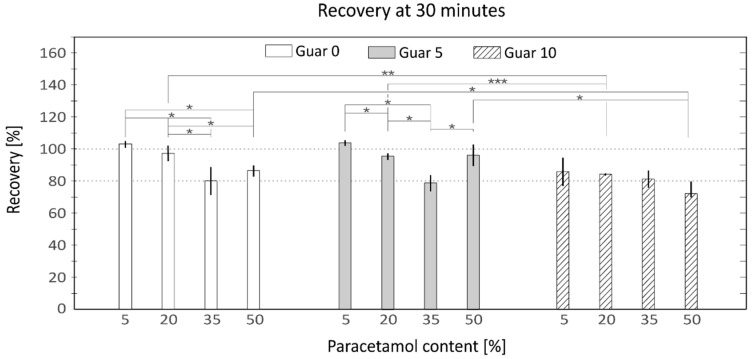
Recovery percentages of each API containing mixture grouped by Guar gum content, white columns 0% wt; Grey columns 5% wt and Striped columns 10% wt of Guar Gum. Horizontal axis represents the percentage of paracetamol, 5, 20, 35 and 50% wt and vertical axis the% rate of recovery measured with HPLC-DAD. (mean ± SD, *n* = 3 * *p* < 0.05, ** *p* < 0.01, *** *p* < 0.001).

**Figure 3 pharmaceutics-13-00907-f003:**
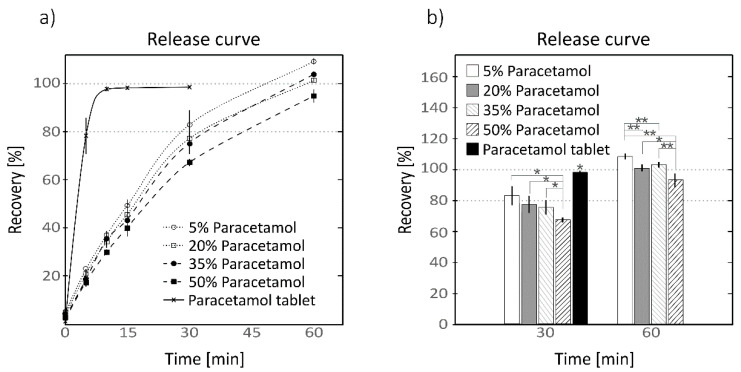
Release curve of extruded mixtures (**a**) containing 5% wt of Guar gum and different proportions of paracetamol and commercial paracetamol tablet at different sampling times; detail (**b**) of recovery values at 30 and 60 min in bar chart showing the statistical significances. Dotted line with hollow circle and solid white columns, mixture with 5% wt of paracetamol; dotted line with hollow squares and grey columns, 15% wt; dashed line with solid circle and grey stripped columns, 35% wt; and dashed line with solid squares and black striped columns, 50% wt. Continuous line with crosses and solid black column, recovery percentage of the paracetamol tablet. Horizontal axis represents the time bar in minutes and vertical axis the % rate of recovery measured with HPLC-DAD (mean ± SD, *n* = 3, * *p* < 0.05, ** *p* < 0.01).

**Figure 4 pharmaceutics-13-00907-f004:**
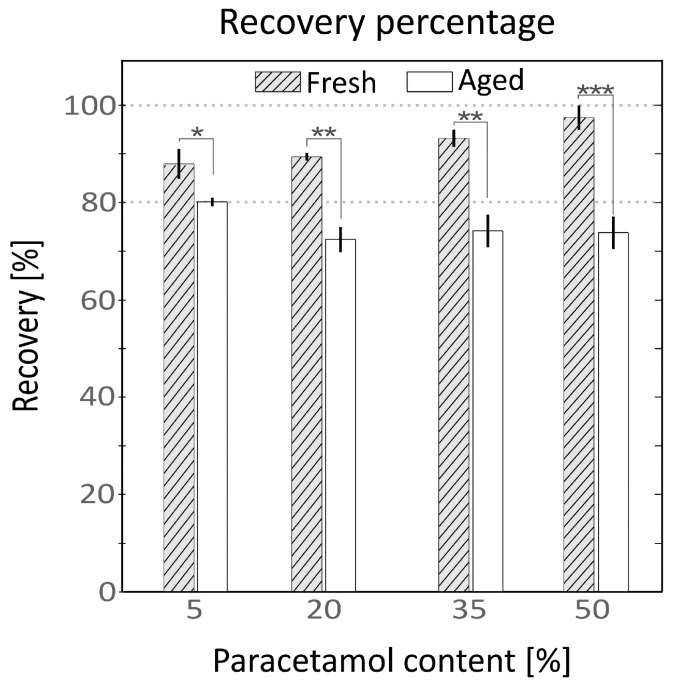
Recovery percentages of samples containing 5% of Guar gum tested in fresh condition (grey stripped) and aged for 6 weeks at room temperature (solid white). Horizontal axis represents the API percentage and vertical axis the % rate of recovery measured with HPLC-DAD (mean ± SD, *n* = 3, * *p* < 0.05, ** *p* < 0.01, *** *p* < 0.001).

**Figure 5 pharmaceutics-13-00907-f005:**
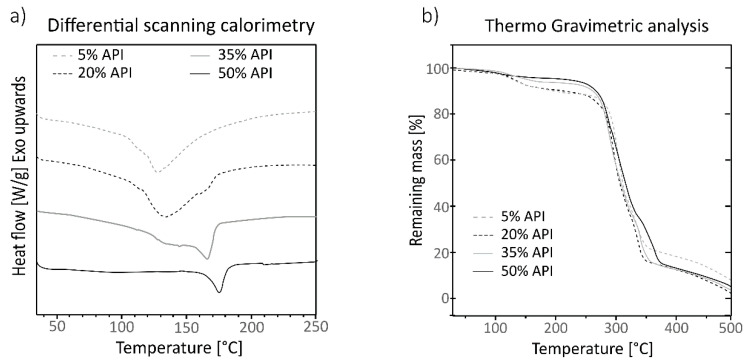
Differential Scanning Calorimetry (**a**) and Thermo Gravimetric Analysis (**b**) of samples containing 5% wt of Guar gum and different proportions of paracetamol, 5% wt dotted grey line, 20% wt dotted black line, 35% wt solid grey line and 50% wt solid black line. The horizontal axis represents the temperature in °C and the vertical axis the heat flow in (W/g) for the graph in the left panel and the percentage of remaining mass (%) in the right panel.

**Figure 6 pharmaceutics-13-00907-f006:**
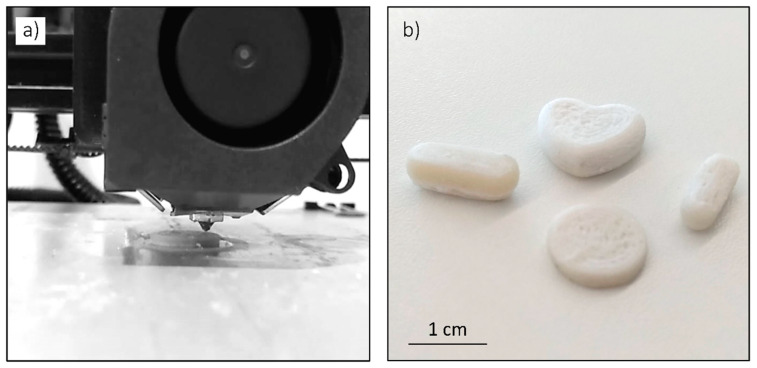
Detail of the 3D printer head printing a 3DP pill (**a**) and some examples of printed pills with different geometries and sizes (**b**).

**Table 1 pharmaceutics-13-00907-t001:** Identification and proportions of paracetamol, starch, HPC, Guar gum, added ultrapure water and calculated total moisture of each sample.

Sample	Paracetamol (%)	Starch (%)	HPC (%)	Guar Gum (%)	Added Water (%)	Total Moisture (%)
FG05702500A	5	70	25	0	2.1	9.1
FG20552500A	20	55	25	0	1.65	7.15
FG35402500A	35	40	25	0	1.2	5.2
FG50252500A	50	25	25	0	0.75	3.25
FG05652505A	5	65	25	5	1.95	8.75
FG20502505A	20	50	25	5	1.5	6.8
FG35352505A	35	35	25	5	1.05	4.85
FG50202505A	50	20	25	5	0.6	2.9
FG05602510A	5	60	25	10	1.8	8.4
FG20452510A	20	45	25	10	1.35	6.45
FG35302510A	35	30	25	10	0.9	4.5
FG50152510A	50	15	25	10	0.45	2.55

## Data Availability

The data presented in this study is contained within the article or the Appendix A.

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
