# Peer review of "Direct Powder Extrusion of Paracetamol Loaded Mixtures for 3D Printed Pharmaceutics for Personalized Medicine via Low Temperature Thermal Processing"

_pharmaceutics, 2021, doi:10.3390/pharmaceutics13060907_

Round 1

Reviewer 1 Report

The authors investigated the powder form of the drug and excipient at low extrusion temperature for the use of modified 3D printers. Potato starch and hydroxypropyl cellulose based mixtures loaded with paracetamol up to 50% in weight were processed by hot melt extrusion at 85 °C. The Guar 76 gum (Gg) is used as binder and disintegrant in solid dosages.

However, the scientific mechanism and experimental data were very limited to prove the goal as compared with other published papers. The manuscript was also written in a premature state, having a lot of errors and incomplete sentences.

Additional comments

  1. The key to extrude the powder samples was to add Ultrapure water, in 3% wt ratio was unclear how water was reliably controlled during the extrusion process.
  2. The current method appears to be case sensitive, giving unreliable drug recovery. It was also questioned why the samples aged for 6 weeks showed a reduction in the recovery up to 20% related to the loss of moisture.
  3. Line 134-135: There was a unclear sentence.

“Error! Reference source 134 not found. summarizes the prepared samples and their composition”.

  1. Line 143: Incomplete sentence.

Extruder temperature 85 °C and 20 rpm.

  1. 2.5, and 2.5.1 part had the same title
  2. Figure 2: Two data was inconsistent. The right graph is redundant.

Reviewer 2 Report

Dear Editor,

Thanks for giving me the chance to review the present work.

The work fits well with the journal purpose as it proposes the use of 3D printing and extrusion process as a tool for medication personalisation. Overall, the reference format needs to be reviewed, and there are some typos. The text needs to be uninformed; for example, sometimes a group of words are capitalised and others not. I do recommend it to process further into the submission process after minor revisions listed below.

Abstract

The abstract needs a sentence to introduce the field and highlight the knowledge gap it intends to fill. The aim is not clearly stated. Please address accordingly.

  • Line 16: please define a "regular" 3D printer as there are different techniques available. Moreover, please do consider adding an extension of the abbreviation "3D".

Introduction

Overall, this is a comprehensive introduction with an excellent supporting background. Please address a few comments reported below.

  • Line 52: No all FDM techniques cause filament formation by an extrusion process. Recently it was investigated the drug impregnation method to improve filament printing. Please add such information to this section.
  • Please add a sentence that identifies the knowledge gap intended to be filled
  • The overall aim is not precise. Please rephrase.

Materials and Methods

  • Line 89: purchased by
  • HSN, PVDF: please state in full
  • Line134: "Error! Reference source not found."
  • How many repetitions of DSC analysis was conducted for each sample?

Result

  • Figure 1: please add a label with the name of each extruded mixtures using the same name in Table 1 or add this information into the caption.
  • Please uniform the text in term of capitals and references.
  • Please justify the text in Figure 2 caption
  • Figure 2: please add the same display for all the p values representation for more consistency
  • Figure numeration is not correct as Fig 2 repeats twice
  • Figure 3 is displayed in the text, but it is introduced into the next in the discussion. Please amend accordingly. Overall, this image should fit into section 3.2.3
  • Section 3.3: the author refers to structure difference, but the chemical structure could be assessed via NMR. Could you please specify?
  • Figure 4: DSC, could you please add the curve of the drug alone?
  • DSC: Please add the excipients' thermal properties (maybe as a table) to highlight that the endothermic peak observed does refer to the drug instead of the excipient. Thus, the drug thermograph needs to be added
  • Figure 4: please address the typos
  • Did you evaluate the release of the drug from the 3D printed tablet?

Discussion

This is a good section with a good explanation of the finding. Please do address some problems that occurred with the references.

Conclusion

A good section, but please use the same abbreviation used in the main text.

Reviewer 3 Report

In this paper, hot melt extrusion was used to print paracetamol with different combinations of potato starch and HPC excipients.  Overall, I thought I agree that 3D printing is an important emerging area in pharmaceutical development of interest to your readership, and the paper was clear, well written and thorough with respect to characterizing the resulting materials.  I felt that more details needed to be provided on the printing behavior and settings, and the impact of these on the behavior, but when these comments are addressed I feel the paper will be ready for publication.

-The authors provide only limited information on the print settings used, limiting the ability of this work to be duplicated elsewhere.  Notably, I’d like to know the build plate temperature and surface material, the layer thickness and print speed, all of which are crucial to repeating the process here.  It also was unclear why they listed the extrusion temperature as in the ’90 to 120°C range’; this is a sizeable range.  Which samples were printed at what temperatures? 

-I would have liked to have been a demonstration of a more complicated 3D geometry printed using this technique, to show that it is possible to print more complex structures with this material.   The examples in Fig. 6 are all more or less 2D geometries.

-Extrusion technologies tend to lead to relatively porous parts compared to alternative fabrication approaches.  Have the authors characterized the internal structure of the resulting prints, and what would porosity do to the drug release characteristics?

-In the review version of the text, a number of the references had not been linked properly leading to periodic ‘Error! Reference source not found’ errors in the text that need to be resolved.

Round 2

Reviewer 1 Report

This paper was improved but it still requires more scientific mechanism and experimental data to prove the goal as compared with other published papers.

  1. What was the selection guidelines of formulations, based on the different proportions of different proportions of starch and HPC +/- guan gum based mixtures with acetaminophen (paracetamol) as drug model?
  2. Authors said that the samples with moisture level kept at 3 %wt of water/starch ratio turned out to be easily extrudable at low temperature (85 °C). What was the experimental outcomes if the temperature was varied. What is the criterion for the optimum temperature of 85 degrees?
  3. The key to extrude the powder samples was to add Ultrapure water, in 3% wt ratio was unclear how water was reliably controlled during the extrusion process. Give more scientific reasons.

2.5. Release Studies

This part was recommended to be mover to somewhere.

 “A 0.5 mg/mL standard solution of paracetamol in 0.1 N hydrochloric acid was prepared for sample quantification. The sample concentrations were calculated by using the response factor (RF) obtained from this standard. The acceptance criterion for each analytical batch was coefficient of variation (CV) ≤ 5%.

Round 3

Reviewer 1 Report

The present paper studies the suitability of different formulations, based on starch and HPC with different proportions of different proportions of starch and HPC based mixtures with acetaminophen (Paracetamol) as drug model, to be processed at low temperature by hot melt extrusion, searching the compliance to be printed in a FDM 3D printer.

However, this paper is not innovative in the design of personalized pills using 3D printing technology. There are numerous related papers similar to this paper mentioning xanthan gum and FDM 3D printer. This appeared to be a case study, just varying formulation without showing scientific mechanism and new findings. Most of all, the authors must show a scientific proof that API did not undergo thermal degradation at the tested low temperature. (When maintained under dry conditions, the API is very stable at room temperature. However, at elevated temperatures and in the presence of trace moisture, acetaminophen degrades more rapidly to p-aminophenol, which subsequently undergoes additional oxidative changes). Due to the lack of science and insufficient evidence, I must reject this paper.

Mahdiyar Shahbazi* and Henry Jäger* A Current Status in the Utilization of Biobased Polymers for 3D Printing Process: A Systematic Review of the Materials, Processes, and Challenges, CS Appl. Bio Mater. 2021, 4, 1, 325–369

Li et al. / Review of 3D printable hydrogels and construct, Materials and Design 159 (2018) 20–38